# Comprehensive Review on Wearable Sweat-Glucose Sensors for Continuous Glucose Monitoring

**DOI:** 10.3390/s22020638

**Published:** 2022-01-14

**Authors:** Hima Zafar, Asma Channa, Varun Jeoti, Goran M. Stojanović

**Affiliations:** 1Faculty of Technical Sciences, University of Novi Sad, T. Dositeja Obradovića 6, 21000 Novi Sad, Serbia; varunjeoti@uns.ac.rs (V.J.); sgoran@uns.ac.rs (G.M.S.); 2Computer Science Department, University Politehnica of Bucharest, 060042 Bucharest, Romania; asma.channa@stud.acs.upb.ro; 3DIIES Department, Mediterranea University of Reggio Calabria, 89100 Reggio Calabria, Italy

**Keywords:** biosensor, diabetes, hypoglycemia, sweat based sensing, wearable electronics, non-invasive glucose monitoring

## Abstract

The incidence of diabetes is increasing at an alarming rate, and regular glucose monitoring is critical in order to manage diabetes. Currently, glucose in the body is measured by an invasive method of blood sugar testing. Blood glucose (BG) monitoring devices measure the amount of sugar in a small sample of blood, usually drawn from pricking the fingertip, and placed on a disposable test strip. Therefore, there is a need for non-invasive continuous glucose monitoring, which is possible using a sweat sensor-based approach. As sweat sensors have garnered much interest in recent years, this study attempts to summarize recent developments in non-invasive continuous glucose monitoring using sweat sensors based on different approaches with an emphasis on the devices that can potentially be integrated into a wearable platform. Numerous research entities have been developing wearable sensors for continuous blood glucose monitoring, however, there are no commercially viable, non-invasive glucose monitors on the market at the moment. This review article provides the state-of-the-art in sweat glucose monitoring, particularly keeping in sight the prospect of its commercialization. The challenges relating to sweat collection, sweat sample degradation, person to person sweat amount variation, various detection methods, and their glucose detection sensitivity, and also the commercial viability are thoroughly covered.

## 1. Introduction

Wearable and digital technologies are bringing innovations to enable individuals with the ability to monitor their fitness and health conditions regularly and non-invasively. These technologies can measure a wide range of physiological parameters, including heart rate and physical activity, but it is lacking in the capability to quantify biochemical parameters that are necessary for the management of a wide range of pathological health conditions. For example, hypoglycemia in which blood glucose level decreases lower than the normal range is a risk for diabetic patients, particularly after they perform intense exercise [1].

Similarly, diabetes being a chronic illness is characterized by an unusual increase in the level of blood sugar, which eventually causes serious damage to the heart, blood vessels, eyes, and kidneys. According to the World Health Organization (WHO) [2], around 422 million people worldwide suffer from diabetes. Regular blood sugar monitoring is very crucial to manage type 1 or type 2 diabetes. It aids the subject to know how the numbers go up or down when eating different foods, taking medicine, or being physically active. The awareness and ability to monitor blood glucose (BG) has led to a significant improvement in the management of diabetes. Combined with therapy and the right protocols, it will gradually halt the rise in diabetes or hypoglycemia issues among people.

Figure 1 presents the vast field of glucose measurement techniques and distinguishes three different categories: invasive, minimally invasive, and non-invasive approaches. Minimally Invasive (MI) technologies are those that need to extract some form of fluid from the body (i.e., tears, saliva, sweat, and interstitial fluid (ISF)) to measure the glucose concentration through an enzymatic approach. In invasive monitoring, there is direct detection of glucose in the blood. In noninvasive or at least minimally invasive methods, biological fluids are obtained which contain glucose at concentrations that correlate with that in blood. Because of this correlation, these alternative biofluids have become novel analytes of interest for the painless monitoring of glucose in the body.

Present BG tracking techniques, however, are invasive, unpleasant, and painful. Currently, a finger-prick test is a common method to obtain an insight of blood glucose level, which is enzyme-based and analyzed by in vitro methods using test strips and a glucometer [4,5]. The effectiveness of this method depends on strict compliance, which can be affected by time constraints or pain [6]. Additionally, it is not a continuous monitoring process and requires testing multiple times in a day in order to manage elevated glucose levels [7,8], especially after performing exercise, having meals, and dosing insulin. Furthermore, a non-continuous approach can miss periods of hyper- or hypoglycemia [6]. Nonetheless, there are implantable glucose monitoring systems that contribute in offering regular glucose monitoring, but these methods are not recommended for all diabetic patients due to their invasive nature, and some of these approaches have been reported to show inaccuracies up to 21% [9]. Thus, there is high consumer interest for a persistent glucose checking framework that can measure glucose levels without incessant calibration. In all the invasive techniques, the blood remains the most studied body fluid. However, in a non-invasive method, various considerations are needed to safeguard the accuracy and quality of measurements of glucose concentrations from these biofluids such as tears, ISF, saliva, and sweat. Among the major considerations, the glucose level in these biofluids is lower than that in blood. In the case of tears, the interference from impurities is relatively small but there are challenges associated with the energy supply for the glucose sensor on the contact lens to operate autonomously and transmit data wirelessly. In the case of saliva, the analyte is easily collected by spitting, but the large amount of impurities in saliva makes it difficult to isolate the inherent level of glucose from the fluid. In the case of ISF, a novel design for a continuous glucose monitoring system has been proposed, but it requires the subcutaneous injection of a needle, which can be uncomfortable to its prospective users. In comparison, the sweat-based glucose sensors are considered as one of the least intrusive solutions to estimate blood sugar level indirectly. However, certain challenges still exist regarding the use of sweat for sampling, the first being its clinical relevancy. Several well-known journals have already reviewed a number of wearable sweat glucose sensors, technologies, and devices, some of which are included hereafter. Toghill and Compton’s survey of electrochemical glucose sensing methods [10] over the previous decade gave an excellent overview of the many kinds of sensors that have been explored. Raman and infrared spectroscopy have received a lot of interest as non-invasive glucose detection techniques, and accordingly, spectroscopic techniques have increased in popularity [11,12,13]. Jayoung Kim et al. [14], in their review article, examined current developments and difficulties in the development of non-invasive epidermal electro-chemical glucose sensing devices focusing on skin interstitial fluid (ISF) and sweat bioanalytes. How to consistently collect a set amount of sweat before analyzing it was addressed in the review paper by Emma et al. [15]. The challenges of sweat sensing in conventional healthcare facilities have been discussed, outlining the fundamentals of human sweat, its properties and characteristics, sweat gland endocrine modeling, and ending with wearable sweat sensing devices being developed for research and commercialization.

This review article provides a current update on the state-of-art in continuous sweat glucose monitoring and their commercial prospects. We extensively cover both the sweat collection mechanisms and various sweat glucose sensing mechanisms comprising the detection and transduction. We are mindful of the presence of quite low levels of sweat glucose and, hence, the requirements of higher sensitivity. We are also mindful that the sensing is fraught with possibilities of contamination and many other challenges. The challenges relating to sweat collection, sweat sample degradation, person to person variation, glucose detection sensitivity, and commercial viability are all covered. It is seen that electrochemical sensing methods, mature as they are, having been used in BG sensors, show greater potential to be commercialized in coming times. Moreover, in this study, the recent breakthroughs in the development of skin-worn, non-invasive electrochemical glucose biosensors, as well as their prospects and limitations for enhanced glycemic management, are highlighted. Section 2 focuses on the usefulness of the sweat glucose biomarker and provides a historical perspective of sweat glucose sensing; Section 3 explains why sweat is a useful reliable bioanalyte that can be used for several biomarkers and an alternative bioanalyte source of glucose monitoring; Section 4 looks into measurement methods deployed so far for the analyses of sweat, including sensor materials and sensors based on different sweat collection techniques; Section 5 discusses the challenges associated with non-invasive glucose concentration monitoring; Section 6 looks into various efforts that are underway for commercializing sweat glucose monitoring. Finally, the last Section 7 gives concluding remarks and future perspectives that may increase forecast accuracy, according to the collected data in this research article.

## 2. History of Wearable Biosensors

In past years, wearable biosensors have spurred new developments in many innovative technologies in several domains, from environmental to biomedical. Wearable biosensors have the potential to provide continuous, real-time physiological information via dynamic, noninvasive measurements of biochemical markers in biofluids, such as sweat, tears, saliva, and interstitial fluid. These sensors can measure analytes that are extremely crucial to rapidly monitor changing biological fluids in real time. It comprises of an element of a biological recognition layer in the sensors which may interact particularly with a target molecule and a transducer that can convert this interaction into a measurable signal. The presence of a biological element (to be analyzed) and its easy availability under normal conditions are the basis of any biosensor. Sweat-based sensors rely on sweat’s ready availability and easy collection process to facilitate providing the particular bioanalyte for further detection and amplification. In wearable biosensors, numerous known components such as receptors, nucleic acids, cells, and several types of enzymes are used. These are the foundation blocks for creating and characterizing biosensors that use optical, colorimetric, or acoustic sensing principles. The working of these biosensors is explained in Figure 2. Sample containing the target analytes is collected through various mechanisms, which, in case of sweat, may take the form of a microfluidic mechanism. A bio-recognition layer performs the detection of the target analyte in the presence of many. A transducer layer converts the detector response to a measurable response, and subsequently, the result is presented via the readout system, wirelessly elsewhere or locally.

In the beginning, blood glucose was targeted, and bio-recognition of glucose was facilitated through an enzyme-based electrode. In the 1960s, a glucose-oxidase-enzyme-based biosensor was established. Since then, several wearable electrochemical biosensors have been studied [17]. Specifically, in 1962, Clark and Lyons proposed the idea of enzyme-based electrodes, which relied on the following reaction:(1)$$\begin{array}{c} \mathrm{Glucose}+O_{2}\overset{\mathrm{GOx}}{\to }\mathrm{Gluconic}\;\mathrm{Acid}+H_{2}O_{2}\\ H_{2}O_{2}\overset{\mathrm{Pt}}{\to }2H^{+}+O_{2}+2e^{-} \end{array}$$

It involves the GOx catalyzed oxidation of glucose by $O_{2}$ with the production of gluconic acid and $H_{2}O_{2}$. The generated $H_{2}O_{2}$ is oxidized at the Pt (Noble Metal) electrode, and the electron flow induced by this oxidation reaction is proportional to the amount of glucose present in the blood sample. This is considered as first generation glucose sensor and has been widely used for the self-monitoring of BG levels. Later, GOx is immobilized on an amperometric oxygen electrode that measures the oxygen used by the biocatalytic process [16]. In 1963, 40 years after the discovery of insulin, Kadish [18] invented an insulin pump delivering insulin and glucagon (to prevent hypoglycemia). The first commercial subcutaneous insulin pump—the Auto Syringe—was introduced by Dean Kamen [19]. The need for blood glucose testing quickly grew.

In 1965, Ames developed the first blood glucose test strip [20], called Dextrostix, using glucose oxidase. This early strip was for physicians’ laboratories, not for home use. The variation of humidity and the low solubility in biological fluids (known as the “oxygen deficit”) could significantly influence the responses of this first generation glucose sensor performance. To overcome this limitation, the second generation glucose sensors were introduced in the mid-1970s. The concept of patients using BG data at home was contemplated by replacing the sensing layer of the sensor with a non-physiological redox mediator that can transport electrons from the GOx to the surface of the sensing electrode.

In 1971, in the United States, Anton Clemens [21] filed the first Patent for a BG monitor, also known as an Ames Reflectance Meter (AMF), for reliable point-of-care (POC) use in diabetic patients that may be performed at home. This second generation glucose meter (AMF) was used with the Dextrostix, requiring small volume of a blood (approximately 50–100 µL) for known and unknown concentrations of BG by automatically assessing the color changes of the strip. BG was previously calculated from a chart by interpreting the change in color, which was visible. The AMF followed the common Ames Eyetone, which was confined to clinical settings such as medical centers and hospital wards. When the possibility for external BG regulation was established by studies using intravenous glucose measurement and infusion of glucose and insulin [22], by Yellow Spring Instrument (YSI) Company, Yellow Springs, OH, USA purchased Clark’s electrochemical biosensor technology in 1975 and launched the first specialized BG analyzer (YSI Model 23 Analyzer).

Biosensors gained prominence in the late 1980s, when scientists attempted to develop revolutionary technologies that resulted in the development of a new class of glucose sensors known as enzyme-free or third and fourth generation glucose sensors [23] that enable wearable biosensors to monitor a person’s physiological biomarkers in real time. This decade observed the introduction of novel biosensor transduction concepts, including fiber-optic and mass sensitive (piezoelectric) devices, in response to the increasing focus on biotech. In the 1980s, commercial self-testing BG strips based on mediator-based enzyme electrodes were also launched. Efforts to adopt subcutaneous Continuous Glucose Monitoring (CGM) yielded a number of effective devices in the 1990s. Generally, the implanted amperometric biosensors monitor changes in glucose levels in the ISF dynamically and provide constant warnings when glucose levels drop. This non-invasive sensing approach displays the strong affiliation between the ISF and blood glucose levels. Despite the demonstrated benefits of CGM, its adoption was slow.

Due to the advent of nanotechnology in the late 1990s, a wide range of nanomaterial-based biosensors have been developed that make use of the appealing qualities of various nanomaterials, such as silicon and gold nanoparticles, for label-free and amplified biosensors, respectively, starting the fifth generation of the Advanced Wearable Biosensor Platform. A number of distinct DNA biosensors were developed in the late 1990s [24]. This led to significant advancement in biosensor technology over the last few years, paving the path for wearable biosensors. These subcutaneously implantable glucose sensors moved in the early 2000s to commercial wearable biosensors that track the real-time glucose level in the ISF, along with diabetes-relevant trends and patterns. A brief representative history of BG monitoring techniques is shown in Figure 3. As can be seen in the figure, the trend has always been towards noninvasive and continuous glucose monitoring. Today, sweat-based glucose sensing is the top contender there, though wearable optical methods are also very interesting, having similar benefits except for the technological challenges of some underlying mechanisms.

## 3. Sweat as an Alternative Source of Glucose Monitoring

Sweat is considered as one of the crucial bio-liquids useful for non-invasive, continuous monitoring applications because of its particular nature. Sweat is the most freely available source of glucose, having the most sampling sites outside the body, consistent access, simplicity of assortment device placement, and availability of physiologically significant electrolytes and metabolites. In human subjects, sweat glucose (SG) has been successfully measured and reported in [25,26]. The association between SG and BG is also explicitly demonstrated [27,28]. Since sweat reaction occurs quickly and the sweat gland is highly vascularized, glucose levels inside the body can be estimated from sweat samples [29]. The concentration of glucose in human sweat is from 0.06–0.2 mM and corresponds to 3.3–17.3 mM in BG [29,30]. However, significant challenges remain in obtaining accurate sweat glucose data, such as changes in environmental parameters such as temperature, contamination from skin, sporadic sampling without iontophoretic incitement, low production rate, and the mixing of old samples with the new samples. Despite the good interaction, glucose level observation in sweat is extremely difficult in view of its low concentration (~100 times lower than BG), which therefore needs highly sensitive devices.

The growing demand of wearable sensors has helped researchers to develop an insight into various technical issues. A device named SwEatch was developed for sodium investigation in sweat and was manufactured utilizing 3D printing techniques by analysts in [31]. A similar device could be handily adjusted for glucose detecting in sweat by integrating a glucose sensor into the platform. Researchers in [32] manufactured sweat-sensing patches which can animate perspiration creation and detect analyte (sweat) concentrations wirelessly using a smartphone. Several other studies demonstrate sweat glucose monitoring systems using a patch type wearable platforms as illustrated in Table 1. Included in the table are some studies for comparison’s sake that are minimally invasive [33,34], making use of ISF as biofluid for glucose detection.

Sweat measurement is currently being used in applications such as the management of certain sicknesses, the prevention of alcohol dependence, monitoring of athletic training and recovery [39], etc. Observing glucose levels is important for managing fatigue levels in sports persons [40]. There are frequent users of advanced fitness wearables with built-in sweat sensors that aid in the detection of dehydration and water levels in the body. In developing nations, there is a rise in the adoption of wearable technology by people for patient care, early disease diagnosis, and point-of-care (POC) health monitoring. Consequently, sweat sensor companies are now selling hydration sensors to athletes and sports players to help them track their hydration needs [41]. Sports players and athletes are looking forward to a market with devices that sense and generate data for each body fluid to monitor various conditions. For example, cortisol sensors [42] are now used particularly for a few days to help measure human stress levels [43].

Although these research works achieved promising results in SG sensing accuracy, there are a few obstacles to overcome, such as real-time signal correction, long-term stability for continuous monitoring, and also the reproducibility between sensors and various patients and customized wireless electronics. The possible options for direct glucose monitoring include body fluids such as sweat, saliva [44], and tears [45]. The glucose density is 1 to 10 per cent of the blood density in these fluids. The long-term view is to develop sensors for integration in wearables such as clothing [46,47], bracelets [48,49,50], patches [51,52], and tattoos [39,41,53], which can sample a number of body indicators continuously. Some examples are given in Figure 4.

It is important to have a thorough understanding of sweat gland features and functionality before determining the most appropriate location for these sweat glucose monitoring sensors to be installed. Sweat glands are present all over the body, but are most numerous on the forehead, the armpits, the palms and the soles of the feet, as shown in Figure 5.

There are millions of eccrine glands [64] that are distributed across human skin and secrete liters of sweat per day. Eccrine sweat glands are present in several areas throughout the body with densities of more than 100 glands/cm${}^{2}$, allowing for a vast array of viable sampling sites. As a result, the argument for utilizing a wearable device to sample and detect sweat is clear. The main advantages and challenges of eccrine sweat are discussed further, and the issues will be quickly described as technological (perhaps directly solvable with enough effort) or fundamental (only indirect ways can be solved).

### 3.1. Benefits of Noninvasive Sweat Access to Bio Fluid

The easy availability of sweat on the skin allows for noninvasive and continuous access and sensing of a bioanalyte that contains health markers for a large number of conditions. One such technique is shown in Figure 6.

Unlike other unobtrusively accessible biofluids such as tears, urine, and saliva, sweat is more suitable for health monitoring. Tears can only be sampled once in a while, urine cannot be continuously accessed, and saliva cannot be used. Furthermore, saliva-based sensing may not be particularly accurate or very reliable as it is affected by the last meal the person had eaten [66]. Sweat-based sensing, however, suffers from often unknown correlation between the biomarkers present in sweat and those present in blood, considered the gold standard. While precise sweat-based detection is a challenging task, it is considered as an ideal candidate for continuous or semi-continuous monitoring over a prolonged period. This information can directly lead to detecting several pathologies. These sweat sensors can be placed in close proximity to sweat-generating sites, which allows for a fully wearable device with minimal sample degradation [67,68]. An analyte such as glucose is also present in the sweat, so using appropriate sweat collection method every several minutes, one can generate useful information about the level of glucose in the system.

### 3.2. Key Challenges in Using Sweat for Sensing

Despite the many benefits, sweat sensing requires huge efforts to address the following key challenges and in establishing a reliable correlation to gold standard BG measurements.

#### 3.2.1. Exposure to Contaminants via the Skin

The skin may be contaminated by dead skin cells, sebum, analytes diffusing through the stratum carneum, and the condensate from trans-epidermal water loss, among other things. These issues must be taken into consideration while creating a sweat sensor, and special attention must be paid to signal analysis.

#### 3.2.2. Quantity of Sweat Readily Available

For some patients, their sweat rate is in sub nano-liters per minute for each eccrine gland while they are in a resting condition. Small sampled quantities need to be rapidly moved between the collecting point and the sensing location via advanced microfluidics. In addition, the body’s temperature-regulating systems allows tiny skin droplets to evaporate quickly. Measures must be made in such a way that the microfluidic design is able to reduce evaporation, assure that sweat reaches the sensors, while avoiding the blockage of dried sweat components obstructing reliable and repeatable flow.

#### 3.2.3. Deviations in Results Because of pH Differences

Changes in pH may affect sensor results, although this depends on the sensing technology used. Skin impurities, chronic conditions, or secreted components may all cause a pH shift. As a result, sensors need to be able to withstand large variations in operation without losing accuracy. Sweat sensors often need the addition of a pH sensor for calibration considerations.

#### 3.2.4. Sweat Glands Periodic Activation

Sweat glands only generate sweat at certain times of the day or night. This must be considered while coming up with an appropriate collecting technique. Additionally, this has an impact on the resulting temporal resolution. This means that sampling within minutes may not be feasible since no sweat is produced when an eccrine gland is active for 30 s and then becomes in-active for 150 s [69].

#### 3.2.5. Sampling Variability within and between People

There may be significant differences between individuals in terms of skin topography, sweat rate per gland, and the number of active sweat glands, as well as between measurement locations on a same person. Sweating output varies widely across individuals for a variety of reasons. In terms of inter-subject variability, physical development [70,71], hydration [72], diet [73], and adaptation to the new environment [74,75] are the most important factors. The time of day [76], the area of the body being sampled [73,77,78], and the technique of sweat induction [62] are major variables for inter-subject variability. Due to these factors, parameter estimation is complicated, and correlations between sweat and blood composition indicators may be disrupted. Because of this, procedures for sweat-based sensing may need more frequent measurements at various time intervals to track changes over the course of the measurement and statistical adjustments.

## 4. Insight into Non-Invasive Sweat Glucose Sensing Technologies

In this section, we examine non-invasive sweat glucose sensors from the points of view of materials, mechanism of sweat collection techniques, methods of sensing, and device integration in order to provide insights into the technological challenges that sweat glucose monitoring systems are currently undergoing. Figure 7 highlights the key aspects in the creation of sweat glucose monitoring devices. Given the increased interest in wearable sweat glucose monitoring, we first review the most recent advances in noninvasive methods which need to be addressed based on developed sensors.

### 4.1. Bio-Recognition—Enzymes, Electrodes, and Non-Enzymatic Approaches in Sweat Glucose Sensors

Non-biological catalysts have recently gained a lot of interest for glucose detection since, they can successfully solve all the limitations of first- to third-generation wearable glucose sensors as discussed previously in Section 2. The new technology resulted in the development of a new class of glucose sensors known as enzyme-free or fourth-generation glucose sensors. Various conducting materials have been used to develop the electrodes of the wearable sweat glucose sensor or detecting layers on the basis of sensor type.

#### 4.1.1. Enzymatic and Non-Enzymatic Electrodes Based on Electrochemical Sensors

The electrochemical sweat glucose sensor uses certain enzymes as receptors for detection. The electrochemical sweat glucose sensors use certain enzymes as receptors for detection. Since 1962, when Clark and Lyons proposed the idea of enzyme electrodes and used GOx, much work has been done, and lately, nanomaterials have been added to enzymatic glucose sensors to speed up electron flow. To date, over 300 types of nanomaterials were found to possess intrinsic enzyme-like activity [79]. For the optimized performance, different nanoenzymatic material can be used depending on the target glucose concentrations in the sweat using the modified electrodes, as shown in Figure 8. Metal nanoparticles, nanostructured metal oxides or metal sulfides, conductive polymers [80], carbon nanotubes [81], and graphene [82] are examples of nanomaterials. For example, a gold nanoparticle (AuNPs)-functionalized ZnO nanostructure on a glassy carbon electrode (GCE [83] may be used as a glucose sensor immobilization matrix [84]. Despite the fact that enzymatic glucose sensors are selective and sensitive to glucose, enzymes are highly costly, fragile, and prone to breakdown when temperature or pH levels are high [85].

In non-enzymatic glucose sensors, the electrodes or the modification materials on the electrode function as electrocatalysts instead of enzymes. However, the production of a structured electrode requires an additional fabrication process, and thus the porous working electrode is commercially feasible only for high-precision devices with which a significantly low target range of glucose must be measured. The enzyme is chemically crosslinked with hydrogels (e.g., chitosan and gelatin), nanomaterials (e.g., carbon nanotubes and graphenes), and other stabilizers (e.g., bovine serum albumin) for stable immobilization.

Single-atomic site catalysts (SASCs), [86] containing exclusively isolated active metal sites, have aroused wide attention due to their specific activity and maximized atomic utilization. Among them, Fe–N–C-based SASCs [87] have a large amount of Fe–N*x* active sites, which can mimic the structure of penta-coordinate heme iron systems in heme enzymes. Due to these unique features, Fe–N–C-based SASCs can enhance peroxidase-like catalytic activities and open new avenues to substitute the natural enzymes in the biosensing field. It has a great potential in substituting natural enzymes for various biomedical applications, such as electrochemical sensors, immunoassays and DNA assays, and wearable biochemical sensors.

The layer of electrocatalytic mediator also plays a role in determining selectivity and sensitivity. The mediator should selectively react with the byproducts after the enzyme specifically oxidizes or dehydrogenizes glucose. Considering the target glucose concentration, an appropriate amount of mediator should be functionalized.

#### 4.1.2. Enzyme-Free Optical Glucose Sensors

Glucose detection using fluorescence-based sensors has made significant development in the previous decade as a viable alternative to electrochemistry. Sensors based on fluorescence-based optical sensing employ light frequencies that vary depending on the concentration of glucose molecules in order to establish the interaction of the fluorescent nanoparticles or semiconductors with glucose molecules. When it comes to how the fluorescence sensors work, there are a number of different approaches that may be used. These include quenching, ratiometric [88], RET [89], and polyethylene terephthalate (PET) [90]. A one-step hydrothermal approach devised by Mai et al. [91] resulted in a vertically aligned ZnO nanotube (NT) modified circuit board substrate. By quenching the photoluminescence (PL) of ZnO NTs, the sensor was able to measure glucose levels. For the most part, the photo-excited electrons in ZnO NTs’ conduction band are recombined radioactively with the emission of photon energy upon illumination with UV light in ZnONTs. UV radiation acts as a catalyst to oxidize glucose molecules, which results in fluorescence quenching and a decrease in fluorescence intensity.

#### 4.1.3. Signal Amplification of Nanomaterials

Signal amplification based on biofunctional nanomaterials has recently attracted considerable attention. Since some nanomaterials have enzyme-like activity, they are usually called nanozymes. Over the past decade, nanozymes have proven to be excellent providers of high-performance and ultra-sensitive biosensors, including colorimetric, fluorometric, chemiluminescent, surface-enhanced Raman scattering, and electrochemical assays. Due to their oxidase-like activity, metallic, metaloxide, and metal-organic nanoparticles are used for colorimetric and fluorimetric sensing of H_2_O_2_, O_2_ and H_2_S, glucose, ascorbic acid, cysteine, GSH, and other bio-thiols [92]. These nonyzymes are used to interact with GOx for high-sensitive glucose sensing by signal amplification. There are two possible theories to understand the signal amplification phenomenon in nanomaterial for the detection of glucose in sweat: activated chemisorption and the incipient hydrous oxide adatom mediator model (IHOAM) [93]. Glucose is anchored to the electrode surface (catalytic surface) by establishing bonds with the empty d-orbitals of the catalyst, which is often a transition metal, according to the first model.

In the second model, adatoms are used which are metastable and have high energy. They refer to the surface atoms on the metal electrode located on the defect sites or ledges and having low coordination number. These species are oxidized in lower voltages. The resulting product upon oxidation is the hydrous oxide species, which has lower coordination number than conventional OH. This model suggests that many electrocatalytic oxidation processes of organic molecules are mediated with the incipient hydrous oxide. The catalyst serves a dual purpose in this case: By chemisorption [94], the reduced form of the catalyst activates the reactant (glucose), and the oxidized form helps the insertion of oxygen or loss of hydrogen [95]. The electrocatalytic oxidation of glucose to gluconolactone is mediated by hydroxide anions chemisorbed on the metal’s reductive sites, generating MOH${}_{ads}$ species. Both models can be used to describe how noble metal electrodes such as Pt and Au catalyze reactions. They, on the other hand, fail to explain glucose oxidation on transition metals such as Ni, Co, Cu, or metal-oxides [96]. Instead, under anodic bias, the metal-oxide with a lower oxidation number will undergo an additional oxidation process and attain a higher oxidation number. With a larger oxidative number, this metal-oxide may generate OH${}_{ads}$ species bonded to the surface. Organic species near to the surface may be oxidized by OH on the surface. Many metal-oxide or transition-metal-based sensors are utilized at high pH values because it is easier to produce OH${}_{ads}$ under alkaline circumstances.

### 4.2. Sweat Collection and Methods

The first step in developing a wearable sweat sensor is collecting the test sample, which is eccrine sweat. Once the sweat glands have produced this sweat, it is transported to the sensor, where it can be measured for analyte concentration. These sensors can be divided into three main categories according to sweat collection techniques, as shown in Figure 9: (1) using microfluidic architectures such as channels and chambers to collect sweat as shown in Figure 7c, (2) using a wicking material to draw sweat from the skin into a collection device, and (3) using a cavity in the device or the skin’s topography to create a volume for sweat collection.

#### 4.2.1. Sweat Collection Substrate Materials and Issues

The material used to fabricate these sweat glucose sensors is determined by criteria such as the sensor’s application, availability, overall production cost, and so on. The raw materials that are typically employed in the development of commercially wearable sweat glucose sensors and insulating substrates are as follows: Polydimethylsiloxane (PDMS) [97], Polyethylene terephthalate (PET) [98], Polyethylene naphthalate (PEN) [99], Polyimide (PI) [100], P(VDF-TrFE [101], Parylene, and Polypyrrole [102].

#### 4.2.2. Microfludic

In [103], the authors reported a battery-free, electronic sensing platform which combines chronometric microfluidic platforms with integrated colorimetric assays. A demonstrated system was capable of measuring sweat rate/loss, pH, lactate, glucose, and chloride levels, all at the same time. The essential design concerns and performance qualities are established by systematic research of electronics, microfluidics, and integration methods. Lately, efforts are afoot to utilize skin-interfaced microfluidic stretchable and flexible electronics and mechanics and create new generation of wearable systems with real-time and noninvasive monitoring of sweat biochemistry. In [104], authors developed a wearable sweat colorimetric sensor as depicted in Figure 10. A soft microfluidic device made using low modulus silicone elastomers serves as a platform for data collection, manipulation, storage, and analysis embedded in the skin. The designs and integrated bioassays provide a lighting-independent colorimetric readout of chloride, glucose, pH, and lactate concentrations. Additionally, measures of the temperature of sweat and the dynamics of sweat release are included (rate and total loss).

The recent work in [105] reported a thin, closed microfluidic device that can directly and efficiently harvest sweat from pores on the surface of the skin and by using a colorimetric approach to determine total sweat loss, glucose, pH, lactate, and chloride simultaneously. The microfluidic system consists of embossed relief geometry bottom layer of Polydimethylsiloxane (PDMS), wherein the serpentine channels are filled in with water-responsive chromogenic colorimetric reagents immobilized on the paper. The first and second parameters, lactate and glucose concentration, utilize the enzymatic assay chromogenic reagent (that is, formazan dyes) for the analyses. While a multi-dye technique binding with a 2,4,6-tris(2-pyridyl)-s-triazine (TPTZ)-based chemical reaction is used to track pH and chloride ion. The embedded near field communication (NFC) electronics serves as a means to capture the images and process the data. Authors in [106] discussed significant progress made in microfluidic systems for sweat collection mechanism and analysis by the set of appropriate enzymatic chemistries and colorimetric readout approaches for determining the concentrations of creatinine and urea in sweat.

#### 4.2.3. Absorption Collection

Fabrics are an ideal substrate for wearable sensors since they are in direct contact with the skin and they can absorb the sweat. In addition, the large textile surface gives enough room to incorporate the related electronics. The fabric base offers a wide range of physical and chemical properties, including wool, cotton, and nylon that can be used to integrate chemical sensors into the textiles. The printed textile sensor is durable, with repeated tension bending and stretching. A bandage printed pH sensor for wound surveillance was recently demonstrated in [107]. Similarly, authors in [108] reported a silk-fabric-derived carbon textiles (SilkNCT)-based multiplex sensor array patch mounted on polyethylene terephthalate (PET) for simultaneous detection of six health-related biomarkers (glucose, lactate, AA, UA, Na${}^{+}$, and K${}^{+}$). Few examples of textile sensors utilizing conductive threads have successfully demonstrated the possible wearable sodium sensor for cyst fibrosis (CF) surveillance, as well as a dehydration sensor [63]. Wearable potentiometric sensors for pH, NH${}_{4}{}^{+}$, K${}^{+}$, and Cl${}^{-}$ were also developed by other research groups. They used carbon nanotube-modified yarn and screen printing technologies [109]. Efforts have also been made to improve lightweight plastic wearable sensors and elastomers. Textile-based colorimetric sensors for pH and lactate use three layers of cotton fabric: (1) chitosen, (2) sodium carboxymethyl cellulose, and (3) indicator dye or lactate assay [110], Figure 11. It can measure the lactate level (0–25 mM) and estimate sweat pH (1–4). In sweat, pH is also regarded as a hydration index. Chloride ion concentrations are a marker of cyst fibrosis, and changed levels of electrolytes are equivalent to an excess of sodium ions.

#### 4.2.4. Epidermal Tattoo

Recent studies demonstrated use of electronic tattoos for sensing biomarkers extracted from the skin. For the effective operation of a wearable sensor, it requires the electrode surface and biofluid to be in contact.

The authors in [111] demonstrated a screen-printed electrochemically based tattoo, as shown in Figure 12, which used an elastomeric stamp to print the electrode directly on the skin to facilitate the efficient functioning of a wearable sensor through chemical sensing [112]. The method involved wetting the customized stamp with conductive ink, followed by contact printing of the electrode design on the skin. Later, the same group proposed a scalable, robust mechanism based on temporary tattoos embedded with electrochemical sensors [111]. The manufacturing method includes the screen printing of conductive and insulated inks in tattoo-formed commercial (T3) paper, using finely distributed carbon fibers. This technique has been further employed by De Guzman et al. for measuring the impedance and changes to skin barrier function [113] using sweat containing lactate [114], pH [110], and ammonium [115] that could aid in the management of diseases such as atopic dermatitis and psoriasis. Similarly, in [116], the researchers developed low-cost, 1.5 µm thick medical-tape-based, breathable electronic tattoo sensors with minimized motion and sweat artifacts using skin-conformable devices by incorporating an open-mesh design that avoids the accumulation of sweat between the e-tattoo and the skin, which helps to minimize sweat artifacts. The ultra-thin e-tattoo can measure electrocardiogram (ECC), temperature, and the moisture of the skin. The reported ultra-thick e-tattoo is capable of sensing even while the body is in motion with minimal movement artifacts.

### 4.3. Transducers and Electronics: Sweat Glucose Sensing

Wearable sensing technology represents an alternative non-invasive solution to the classic medical and biochemical approaches that requires the collection of a blood sample using in-depth physical trauma and dangerous infection techniques. In the case of glucose monitoring, which is particularly unavoidable for diabetic patients (for both type 1 and type 2 diabetes), self-monitoring using sweat sensors will provide less intrusive methods. Alternative technologies can be minimally invasive to blood glucose by measuring ISF levels or by using non-invasive techniques without any skin barrier penetration. There are a variety of technologies available for measuring glucose concentrations, each of which makes use of a different chemical or physical methods. Aspects of the underlying sensing methods include the electrochemical method (primarily enzymatic detection of glucose in sweat, tears, and saliva combined with other techniques such as amperometry), optical methods, and electromechanical methods, as shown in Figure 7c, fifth layer. There are a few other other miscellaneous techniques that are catching up such as bioimpedance analysis, as captured in Figure 1.

#### 4.3.1. Real-Time Sweat Monitoring using Electrochemical Methods

Various detection systems have different system requirements and platforms. While optical and other methods have advantages in certain applications, the electrochemical method has shown wide success for fluidic analyses in terms of high precision, selectivity, low reaction times, and easy adaptability to wearable formats. All electro-chemical sweat sensors use three to four electrodes manufactured on flexible substrate. These electrodes are working electrodes, counter electrodes, and reference electrodes and cathodes. Different electrochemical sensing techniques such as potentiometry and chronoamperometry, anodic streaming square wave voltammetry (SWASV), cyclic voltammetry (CV), differential pulse voltammetry (DPV), and electrochemical impedance spectroscopy (EIS) can be used for the analysis. Table 2 shows the summary of electrochemical methods for the detection of sweat in real time.

In the potentiometric technique, the developed electrode potentials undergo observable shifts in concentration with the detection of target dominant ion in sweat with (60 mM Na^+^ for cystic fibrosis patients), such as metabolites (e.g., lactate), electrolytes (e.g., pH, sodium), and cortisol [118,119] with ±3.0% relative standard deviations (RSD), also as reported in [120] with sensitivities of 10.89 µA mM${}^{-1}$ cm${}^{-2}$ and 71.44 mV pH${}^{-1}$ for glucose and pH with mechanical stability 30% and stability for 10 days.

Chronoamperometry is used for the detection of the current produced (using enzyme-based sensing) from the redox reactions of the targeted analyte activated by constantly applied potentials [56,121] by using mediators such as ferric material and Prussian blue in between difficult-to-access redox centers in the enzyme to the electrode as shown in Figure 13. For metabolite sensing, biosensors with enzyme recognition elements are used where the enzyme is tethered using an enzyme immobilization process on working electrode including entrapment, covalent cross-linking, or bonding. To initiate the electron transfer process, the enzymes work as catalyzing a reduction–oxidation reaction between the working electrode and the enzyme. When the product concentration increases due to used electroactive substances, then chronoamperometry is used to read the enzymatic reactions.

As reported by Mengke ku et al. [99], they fabricated a flexible chip using an electrochemical deposition method via a cross-linker poly(ethylene glycol) diglycidylether (PEGDE) on a working electrode as enzyme immobilization using gold nano-pine needles (AuNNs) as the signal amplification strategy, which can monitor real-time glucose and lactate in sweat with low detection limit down to 7 µmol/L${}^{-1}$ and 54 µmol/L${}^{-1}$ up to 4 weeks. Voltammetry analyses are carried out by balancing through the potential ranges of the redox reactions of the target organisms and then calculating the redox current peaks. Specifically, Cyclic Voltammetry (CV) is used to investigate the kinetics of electron transmission and redox processes for the preliminary electro-chemical characterization of the sensor, as well as molecules such as glucose and uric and ascorbic acid in sweat [111,122]. However, Differential Pulse Voltammetry (DPV) is used to detect organic or inorganic organisms by a stimulus attaching to the sensor surface and is also applied for detection of different proteins [108,123,124].

Similarly, Square Wave Anodic Streaming Voltammetry (SWASV) has shown effectiveness for heavy metal identification in particular when combined with active pre-concentration on the sensing surface of the target species in order to enhance the signal [125]. Despite the fact that this method demonstrated a low-detection level, a major problem with voltammetry methods is the overlapping of the redox potential, the presence of active intervening compounds, and the formation of intermetallic degradation-signal compounds [126]. Electrochemical Impedance Spectroscopy (EIS) is used by analyzing dynamic impedances in Nyquist plots to translate bio-affinity binding and can often be used as an alternative detector to voltammetry approaches. In general, EIS needs longer measuring and post-processing times to address measurement uncertainties.

#### 4.3.2. Optical Methods

Several investigations on non-invasive monitoring [130] have been undertaken by researchers, such as optical techniques, which are one of the interesting approaches in CGM. It uses light interaction to measure the change in scattering properties in tissue as a function of glucose concentration. Generally, spectroscopy-based techniques rely on identifying the unique molecular absorption signatures of the target metabolite in the visible or infrared spectrum of the reflected light [131]. The interference of confounding variables can be easily found in optical techniques such as infrared spectroscopy [132], e.g., Near-Infrared (NIR), Mid-Infrared (MIR), Raman Spectroscopy, Optical Coherence Tomography (OCT) [133], and Fluorescence [134]. Additionally, changes in temperature, perspiration, and motion also cause errors, and calibration is required for the technique to produce reliable measurements [135].

Optical methods have a number of advantages for CGM in general. Because most optical techniques are not reliant on reagents, their lifetimes can be significantly longer than those of electrochemical sensors. This is particularly critical when performing minimally invasive or invasive measurements. Even fluorescence sensing systems, which rely on fluorophores, appear to have significantly longer lifetimes than current electrochemical sensors. Because optical sensing does not consume glucose, the glucose concentration in the area surrounding the sensor remains unchanged. Noninvasive optical measurements also have the potential to enable painless CGM with minimal effect on the body, assuming that high measurement accuracy can be achieved. Smaller and more affordable components are constantly being developed, which benefits personalized devices. Each optical technology has distinct advantages and disadvantages.

Although NIRS has been extensively studied for noninvasive CGM, no devices have passed on the market. There are fewer studies on MIRS for glucose measurements compared to NIRS. Some research has shown that non-invasive glucose monitoring systems such as NIRS and MIRS can accurately measure glucose levels in a variety of fluids, including ISF. Fluorescent-labeled CGM systems have made the most progress in comparison to other optical methods. In addition to the Eversense system [136], as shown in Figure 14, several other devices are in the prototype stage. Good fluorescence labeling makes it possible to measure glucose accurately, and the sensors have a long lifespan. With increased investments in body fluid research and development, the usage of these sensors will expand more in the coming years. However, optical methods are likely more costly than electrochemical methods, so there are not as many efforts towards the commercialization of these.

In a nutshell, the optical method’s goal is to estimate blood glucose levels indirectly rather than directly by measuring glucose concentration in the body. Optometry may reach non-invasive conditions, but it cannot be used in clinical or commercial practice since the measured value is not well associated with the real glucose levels and the linear range is restricted, requiring algorithm optimization. Every individual is different from each other, so are his bioanalytes; hence, the existing general fitting method approach based on experimental samples cannot be applied to all of them. Another factor limiting its potential application in the home is the difficulty of detecting glucose, as well as the high needs for detection equipment and considerable interference from background signals.

#### 4.3.3. Electromechanical Methods

The other fundamental sensing methods used in wearable sweat-sensing devices can be categorized into electromechanical techniques. These can be divided into the subcategories of piezoelectric [137], surface acoustic wave (SAW) [138], and photoacoustic spectroscopy [139]. A piezoelectric-material-based crystal or plate resonator is used in piezo-based sensing. The assumption being that the resonator frequency undergoes a proportional change if the piezo material adsorbs the glucose with the help of glucose oxidase. The same applies to SAW-based sensors where the frequency of the operation can reach up to 2.4 GHz, thus improving the sensitivity. The sensor design is also easier, since the device fabrication technology is all planar, and the SAW velocity can be affected and controlled by glucose adsorption on one surface alone. In the photoacoustic approach, on the other hand, a light source is used, and the introduced light is absorbed by the tissue. Thermal expansion occurs as a result of the absorbed optical energy, causing local heating, and ultimately, in turn, the thermal expansion causes a sound wave. A piezoelectric transducer detects the sound wave in a photoacoustic cell (microphone). Open-ended photoacoustic (PA) cells are often used, having the exposed open end in contact with the test specimen. Specific wavelengths that target glucose absorption bands enable the selective detection of glucose. The changes in the mechanical properties of a layer are linked to glucose concentrations by incorporating a boronic-acid-based hydrogel as one of the layers in the stack. The volumetric changes in the hydrogel are measured as changes in the frequency position of a resonance peak of the sound waves. The advantages inherent in such a method would be there being no requirement for an in vivo power supply or on board circuitry. Furthermore, the use of boronic-acid-based hydrogels provides a stable selective and potentially accurate method for sensing glucose concentrations, which might have some advantage over enzyme-based sensing methods, which can suffer from denaturation and degradation.

For the photoacoustic spectroscopy of glucose [140], quantum cascade lasers (QCLs) or Fourier transform infrared (FTIR) sources are most generally utilized since the MIR absorption bands are usually targeted as illustrated in Figure 15. However, the fundamental disadvantage of this arrangement is its low sensitivity for in vivo glucose detection. If the NIR absorption bands of glucose are targeted, NIR light sources may be employed. The challenges in PA spectroscopy for glucose monitoring are partially determined by the targeted glucose absorption bands. For MIR absorption bands, it is necessary to use weak FTIR sources or expensive QCLs, while in NIR bands, the signal becomes much weaker. The scan time of the light source is the primary limiting factor in PA spectroscopy acquisition times.

A number of NIR/MIR spectroscopic non-invasive glucose sensors have proven commercially successful [141]; however, the sensitivity and selectivity of these sensors as well as their correlation with real readings and subsequent algorithms need to be further enhanced.

#### 4.3.4. Miscellaneous Techniques

There are other approaches such as bio-impedance spectroscopy [142], microwave spectroscopy [143], ultrasound [144], and heat propagation [145] which shows strong potentials to be commercialized, probably in coming years. In order to demonstrate the usefulness of any of these techniques, the demonstrators still have to become technologically more complete, they must be tested with a large number of individuals in real-world scenarios, and the measurement instability resulting from human physiological variability must be investigated.

### 4.4. Integration of Wearable Sensor

In sweat-sensing wearable devices, interaction with one or more sensors is performed through wireless communication, thus enabling users a higher range of freedom and flexibility to use these wearable sensors, as shown in Figure 7d. The wireless communication technologies, e.g., Bluetooth, Wi-Fi, and near-field communication, are integrated into the system to supply a fully automated and valuable tool for health monitoring in wearable sweat-monitoring devices. Sensor measurement and data transmission are powered by kinetic and chemical energy generated mostly via the combination of enzyme reactions. After processing the received data in the analog and digital division of the signal conditioning circuit, the data are transferred from the sensor node to the monitoring unit via a router for further analysis. The cost of setup, power consumption, the number of sensor nodes, and the range of transmission are all factors in deciding which communication network to use. Bluetooth has shown to be the most cost-effective option because of its low installation costs, lack of hardware, and high compatibility. That is why much research has been conducted on the implementation of Bluetooth-enabled healthcare systems. Antenna and RF systems embedded in wearables have also been researched as part of the Body Area Network (BAN) [146], where low-powered devices would be surface mounted on the clothing in a permanent location. Off-body, on-body, and in-body BANs are the three types of BAN [147,148]. The benefit of adopting self-powered systems [104,142] is that the enzymatic bio-fuel cells serve as self-powered sensing modules.

## 5. Challenges Opportunities in Wearable Technology for the Analysis in Sweat

In this section, the classic and modified sweat sample methodologies are summarized and emphasized. Traditional sweat analysis is based on extracting perspiration using a whole-body wash down procedure [149] using patches [118], polymer bags/films, macroducts [150], or by any other specialized equipment, facing a number of issues such as unpredictable sweating rate and region, non-real-time monitoring, unavoidable sample evaporation, and contamination. Recent advancements in wearable devices enable sweat collection, storage, and analysis to be integrated into a multifunction platform. Epidermal electro-chemical glucose biosensors are still emerging in the management of diabetes. Despite significant progress made so far, the practical adoption of epidermal monitoring devices that combine precise real-time glucose measurements with long-term stability presents significant hurdles. In particular, large-scale studies are necessary to evaluate the accuracy and reliability of the glycemic devices for people with diabetes.

### 5.1. The Sample Quantities Available Are Extremely Small

At present, the major ways of stimulating sweat generation are intensive exercises or iontophoresis. Sometimes, even with the workout, sweat secretion rates will not surpass 20 nL/min/gland. For disabled men or newborns to obtain sufficient sweat in inactive settings, compared to exercise, iontophoresis stimulation is mostly recommended. However, the current density of the device needs to be adjusted carefully since repetitive application of iontophorous current in the same region might damage the skin. In order to improve sensory performance, ultrasensitive sensors can be developed that function at low sweat levels or extremely efficient procedures for extracting biomarkers.

### 5.2. Reducing Skin-Surface Contamination

It is difficult to exclude the contamination of the skin or the surrounding environment when perspiration is released from the gland, and contact with the skin has a significant influence on the precise reading of sensors. Easy surface cleaning cannot reduce contamination wherein a high level of bacteria is estimated to be 10/cm^2^ in the skin and can cause considerable errors in glucose, protein, and cellular metabolites in sweat in concentrations. One way of solving this problem is to isolate sweat from the surface of the skin by putting a layer of petroleum oil or jelly on the skin. There are several caveats that should be considered while sampling sweat. Ideally, the sensor or sample component should not slide much on the skin, which can allow old and contaminated perspiration to be trapped. To minimize the dead volume between the sensors and the surface of the skin, the position on the skin should preferably remain close in touch, as shown in Figure 16. If the dead volume between the device and the skin can be minimized, this will require less sample volume and also reduce time for obtaining a new sample, increasing the flow-rate across the sensor. This solves the challenge of new sweat mixing with old sweat and contaminating it.

### 5.3. Sensor Shelf Life

According to several studies [151,152], the wearable sweat glucose sensor has a 14-day standard life span before it must be replaced by the end user. It is important to figure out how to extend the sensor’s shelf life. To improve the shelf life of sensors, bioreceptor stability and accuracy should be increased. As a result, the commercialization cost of sweat glucose monitoring sensor may be influenced. Wearable sweat glucose sensors might improve by using artificial enzymes (such as nanozymes) [86] as glucose recognition molecules to achieve reliability.

## 6. Sweat Glucose Sensors Commercial Validation

The rising popularity of wearable sweat glucose monitoring sensors among the people and the healthcare business is as elaborated in Table 3. Geographically, North America has the biggest market share of sweat glucose sensors, as compared to other regions, due to a relatively high sensitivity to health and fitness. Furthermore, rising psychiatric illnesses among young people as a result of stress is another factor driving the market in the region. Similarly, the use of wearables in the Asia Pacific region is predicted to enlarge over the projected period owing to increased investment in the healthcare sector and expanding technology use.

According to the report [154] the global sweat sensor market is predicted to expand to USD 2.59119 million throughout the projection year (2019–2027) with a double numerical increase. The rise of the sweat sensor market is likely to lead to an increased demand for wearable health monitoring systems. Regional bifurcation studies also assess key business tactics such as mergers and acquisitions, affiliations, collaborations, and contracts implemented by these leading market players. There are a few sweat-sensing devices that have been lately developed by many startups and companies that are in the process of entering the market but not yet commercially available. Examples include many, such as BACtrack’s [155] yet to be FDA approved product for transdermal alcohol content monitoring shown in Figure 17a. It is also worth mentioning Alcopro [156], which is trying to commercialize a sweat-based drug testing system designed for testing marijuana, cocaine, opiates, etc., shown in Figure 17b. The device is basically an absorbing pad which is kept in place with an adhesive film and analysed in a laboratory after 7 days. Yet another product is called SCRAM CAM [157], a transdermal ankle bracelet system for continuous alcohol monitoring, shown in Figure 17c. Unlike other systems, this bracelet does not collect sweat; instead, it measures alcohol concentration from vapor releasing from the evaporating sweat from skin. However, the bracelet is equipped to send the information wirelessly to a base station.

A Macroduct Sweat Collection System [158,159] developed by in vitro diagnostics consortium ELITechGroup, shown in Figure 17d, is the only device that is an established method for diagnosing cystic fibrosis from sweat. The system consists of three parts, one that stimulates, another that collects, and the last that analyzes. The collection is performed in a disposable microfluidic plastic device.

### 6.1. Technological Challenges in Commercialization of CGMs

The lack of wearable sweat-sensing devices on the market may be attributed to multiple technological challenges which should be solved. The following need to be considered in order to make these products and the ones which are still in development process to be commercialized:

#### 6.1.1. Large-Scale Sensor Manufacturing

Upscaling for commercial production not only requires a highly viable manufacturing technique but also a relatively cheap cost of production. It still remains a challenge.

#### 6.1.2. Stability of the Sensor

The stability of non-invasive enzymatic sweat glucose sensors may be affected by the breakdown or leaching of the enzyme directly into the surrounding biofluid. False positive results could be generated due to the interferences or delays in signal response, which lowers biosensor accuracy due to the diffusion of the analyte to the sensor surface. Nanoporous enzymatic membranes have been investigated as a possible solution to this problem as they provide good surface area for molecular or ion diffusion and interactions. It is still necessary to develop other recognition methods such as molecular imprinting, peptide chains, and click chemistry, etc., in order to avoid biofouling. The best numbers of stability on sweat-based CGM sensors are below 14 days before biofouling and other issues catch up with sensing.

#### 6.1.3. Reusability and Long-Lastingness

Actually, adhesives applied in many wearable devices may last in the sensor for few days only and slowly degrade over time because of numerous causes, including stratum cornea, skin oils, chemical probes, etc. Therefore, new sensors that are dependent on new materials and manufacturing techniques so as to provide longer reusability and durability are urgently needed.

#### 6.1.4. The Specificity and Sensitivity

In some circumstances, sweat is often diluted, so ultra-sensitive sensors with pre-concentration technologies are necessary. In addition, sometimes contamination and non-specific binding may also cause a signal shift, therefore, the specificity of the sensor is critical and needs to be carefully retained through the lifetime of sensors.

#### 6.1.5. Various Sweat Analyte Extensions

Present sensors are capable of monitoring certain types of electrolytes and sweat metabolites. It is therefore the future need to manufacture wearable multiple sweat analyte sensor.

### 6.2. Calibration

Auto-calibration systems based on large-scale testing should be used in wearable devices, as environmental humidity or temperature, as well as individual divides in diets and in sweat collection areas, easily affect the findings of sensors. These limitations imply that the commercial use of wearable sensors remains an open challenge, which encourages more researchers to work on meeting these challenges of the commercialization of such wearable sensors. We expect to transform biodetection for more personalized and predictive healthcare by using non-invasive wearable sweat devices when these challenges are overcome.

## 7. Concluding Remarks and Future Perspectives

The review herein attempted to capture the state-of-the-art in non-invasive continuous glucose monitoring using sweat. Other non-invasive CGM techniques not using sweat are also briefly touched upon, as they too have potential to become wearable. However, the focus is primarily on sweat-based CGM.

The wearable sweat devices provide an easy and continuous sensing of critical electrolytes and metabolites in sweat by the use of suitably modified absorbent materials, super hydrophobic/super hydrophilic surface, or epidermal microfluidic channels as an effective sampling approach. In recent years, there has been substantial study towards in situ, non-invasive, real-time detection of varying concentrations of sweat analytes. The improvements in manufacturing processes and in the material sciences have permitted the integration of several sensors with on-site circuits for signal pre-processing and wireless data transmission into one mechanically flexible multiplex system. The combination of customized materials has demonstrated that the enhanced surface area and porosity offered, for instance, by nanofibres or nanoparticle-modified electrodes, can improve sensor performance with high sensitivity, low LoD (limit of detection), and wide linear range. Doping with conductive nanoparticles has also proved effective in making non-enzymatic electronic sensors with greater LoDs, sensitivity, and stability compared with enzyme sensors because they have excellent specificity. The presence of various metabolites and electrolytes may supply necessary information for the monitoring of many significant health conditions but needs in vivo validation experiments for meaningful medical applications to check the association of sweat with blood readings. Integrated wearable devices can now induce iontophoresis to sweat and remove the requirement for sweat collection. They can also administer on-demand blood glucose medications for hypoglycemia.

Although many novel sample methods make wearable devices more sensitive, there are various hurdles for future applications with real-time, non-invasive sweat monitoring. The benefits of several sweat sample methods described in this review were presented, but the limitations remain among optical and electromechanical methods. Calibration-free sensors associated with the disease must be introduced to achieve a wider use of devices in integrated multiplexed systems. Wearable sweat devices for clinical applications have not yet been introduced given the present limitations.

In coming days, the trend is towards producing wearables that detect many analytes in sweat simultaneously. Wearable sweat sensors have been recently designed to test analytes of interest using a number of biosensors. Either sweat or other non-invasive biosensors previously mentioned can only track one specific analyte at a time or reliably analyze the physiological situation by sensor calibration mechanisms. Given its scope, simultaneous and multiplex scanning is important for appropriate biomarkers and needs complete device integration to ensure that measurements are correct. Secondly, auto-calibration techniques based on large-scale testing in wearable sweat glucose devices should be implemented as the results of sensors can readily be affected by moisture content or temperature or individual differences in diets and sweat collection regions.

These limitations show that the commercial production of wearable sweat glucose sensors is challenging but offers great promise to inspire more researchers to expedite the marketing of these wearable sensors. It is expected that more commercialization will follow beyond the electrochemically based sensing approach. The common endeavor is to improve biological detection so as to provide better customized, predictive, and real-time healthcare and point-of-care services after addressing these problems with non-invasive wearable sweating technologies.

## Figures and Tables

**Figure 1 sensors-22-00638-f001:**
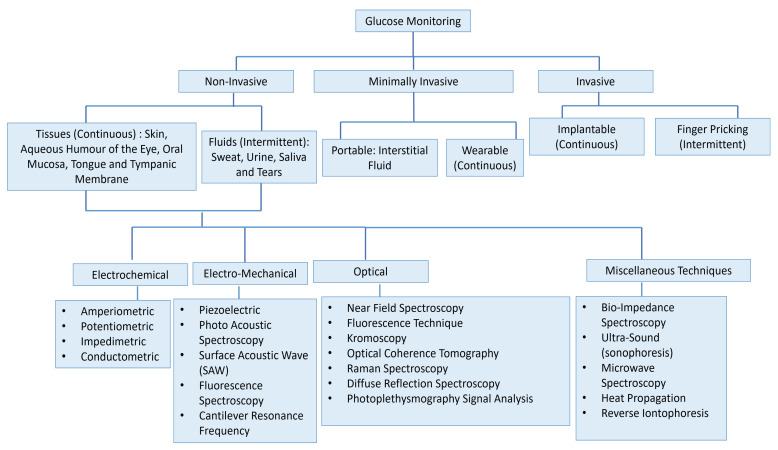
Overview of the numerous approaches for the measuring of glucose for intense insulin therapy. The figure concept adapted from [3].

**Figure 2 sensors-22-00638-f002:**
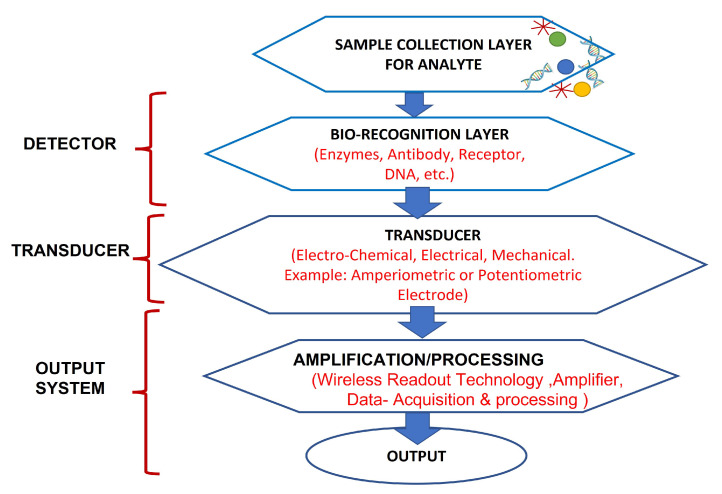
Working of biosensors: A close relationship between certain bio-recognition layers and electrical signal transaction. The figure concept adapted from [16].

**Figure 3 sensors-22-00638-f003:**
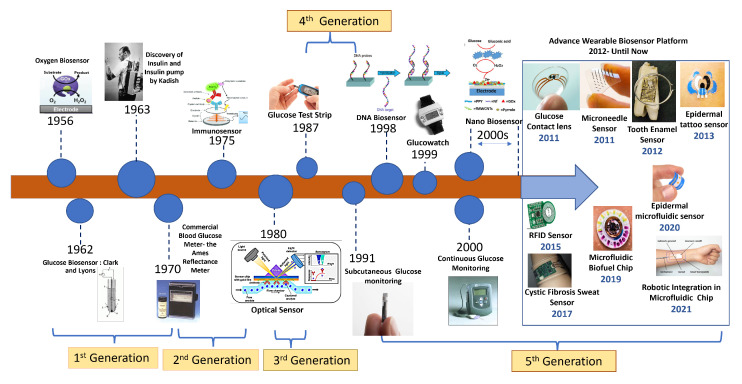
History of biosensor development for wearables from the beginning to the present.

**Figure 4 sensors-22-00638-f004:**
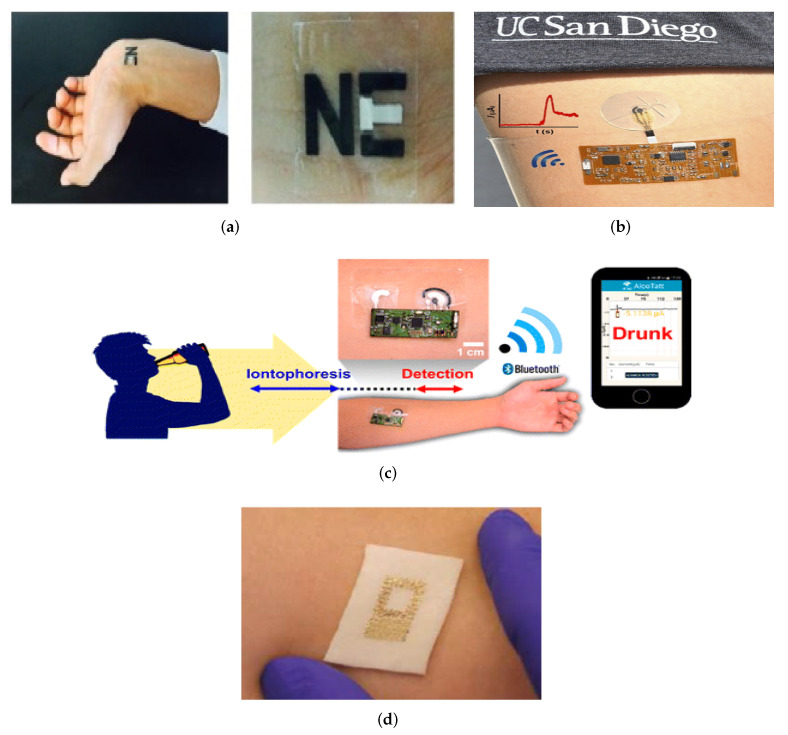
The direct application of sensors to skin, e.g., (**a**) Tattoo [54] (Reused with the permission of Copyright 2019, Elsevier), (**b**) Epidermal Microfluidic Electrochemical Detection System [55] (Reprinted with permission of Copyright© 2017, American Chemical Society), (**c**) Wearable Tattoo-Based Iontophoretic-Biosensing System for detecting Alcohol in Sweat [56] (Reused with permission of Copyright© 2016, American Chemical Society), (**d**) A passive wireless capacitive sensor [50] (Reused with the permission of Copyright 2014, John Wiley and Sons), addresses several difficulties in history. The technology is mechanically compatible with the skin, and all of the examples are important as a first step towards reducing perspiration between the skin and the sensors.

**Figure 5 sensors-22-00638-f005:**
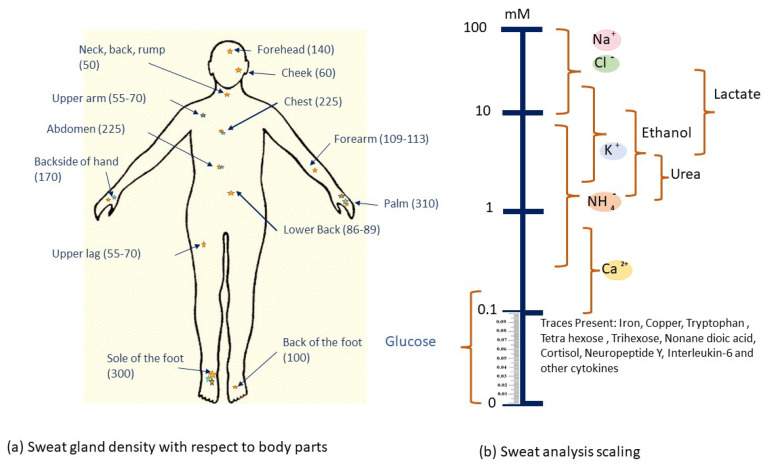
(**a**) The average sweat gland density illustration (glands/cm${}^{2}$) on different areas of the body [57,58]. (**b**) Scale illustrating the approximate range of levels for several sweat analytes reported in [59,60,61,62,63].

**Figure 6 sensors-22-00638-f006:**
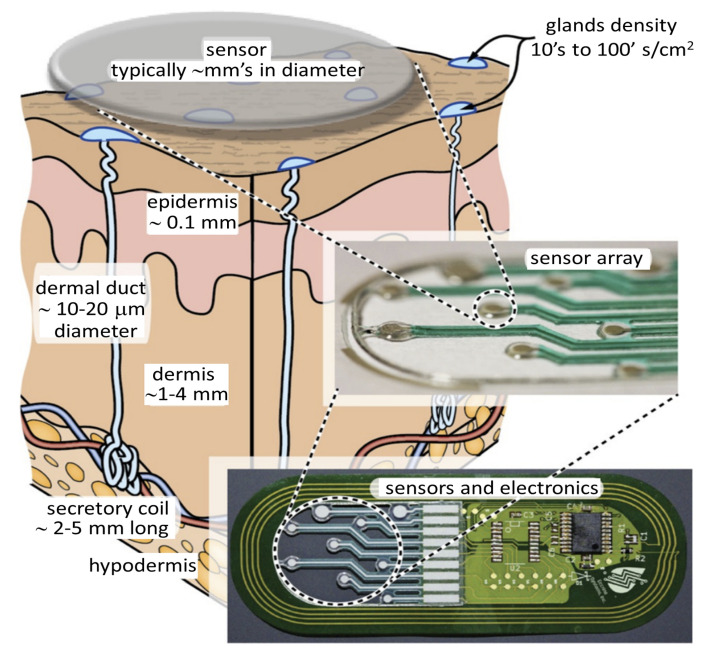
Technique for detecting analytes in sweat as it is released onto skin using multi-analyte RFID patches. The figure concept adapted from [59,65].

**Figure 7 sensors-22-00638-f007:**
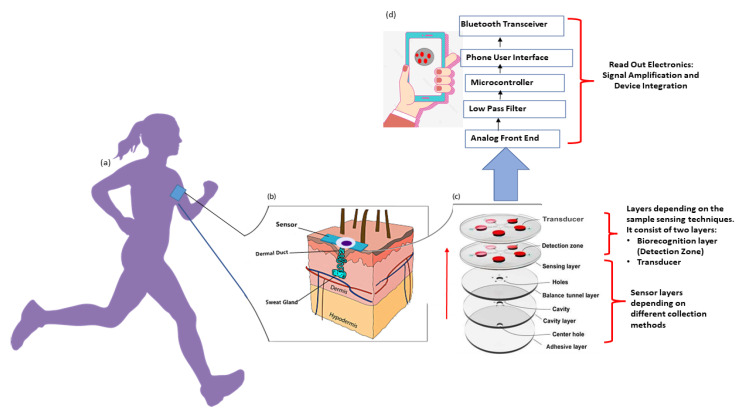
(**a**) A typical commercial non-invasive glucose monitoring (GM) system is depicted. (**b**) The sensing component of the GM system is being implemented, integrated, and placed directly on the skin. A cross-section of the created GM device inserted into the skin is shown. (**c**) Illustration of the sensing layers of non-invasive sensors is shown in larger view; the final layer is dependent on different sensing materials which respond to specific analytes, i.e., sweat, and a protective layer to confine the reaction compartment. (**d**) The integration of a electronic device/system for signal processing and transmission wirelessly.

**Figure 8 sensors-22-00638-f008:**
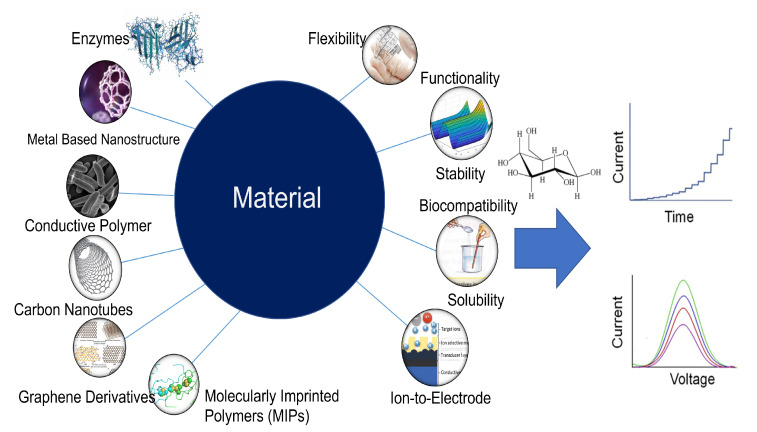
Various sensing materials employed in electrochemical wearable sweat glucose sensors based on their bio-compatibility with catalysts.

**Figure 9 sensors-22-00638-f009:**
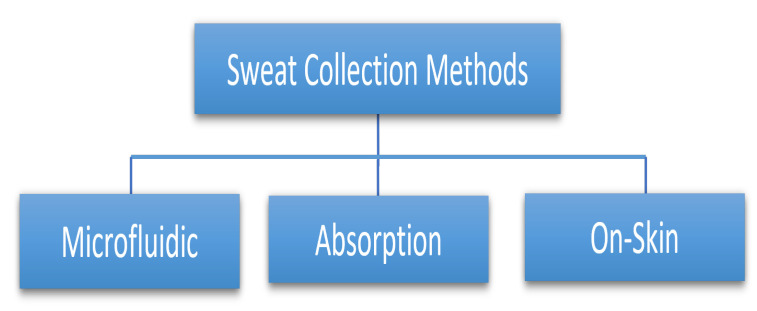
Techniques for sweat collection.

**Figure 10 sensors-22-00638-f010:**
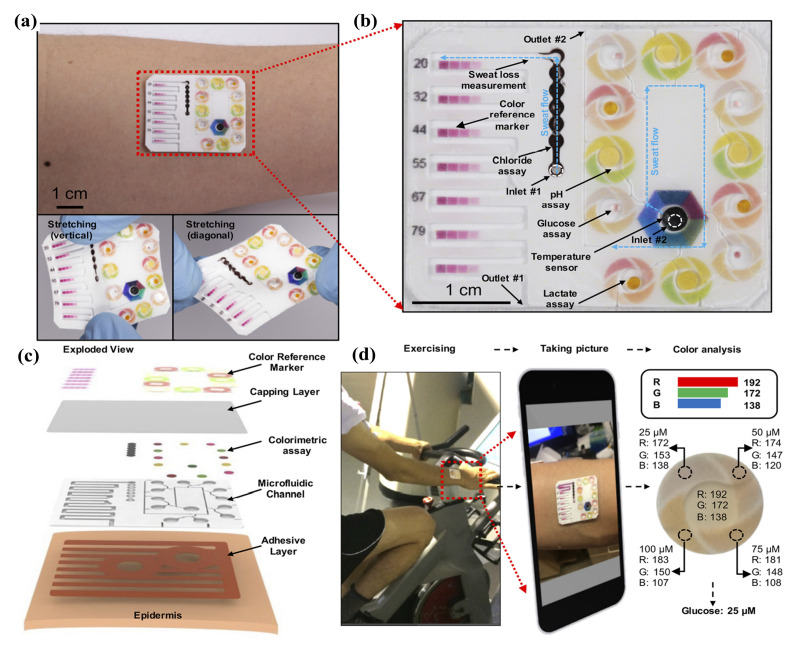
The wearable sweat colorimetric sensor: (**a**) Optical pictures of flexible, soft microfluidic equipment used for colorimetric skin sweat analysis and mechanical bending (bottom left) and twisting (top left) deformation (bottom right); (**b**) Microfluidic channels filled with blue-dyed water from the top; (**c**) A gadget and its interaction with the skin in exploded view; (**d**) The sweat sample collection process and color analysis of the device’s digital pictures. Reprinted with permission from [104]. Copyright 2019 American Chemical Society.

**Figure 11 sensors-22-00638-f011:**
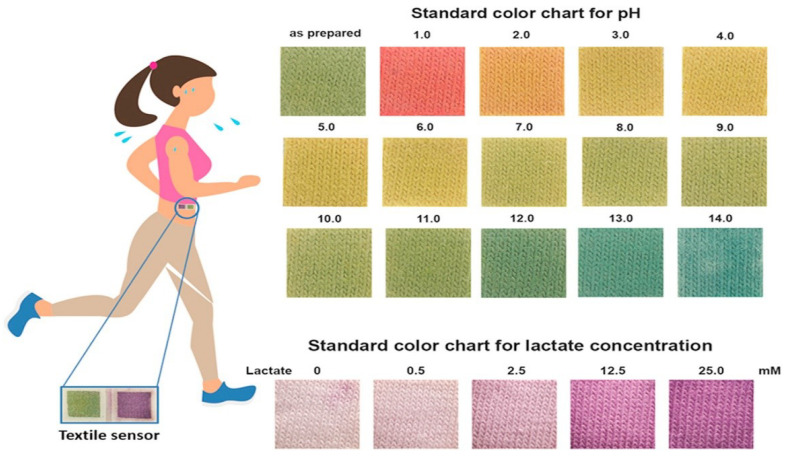
Textile based wearable sweat colorimetric sensor, using knitted cotton fabric material, for simultaneously detecting sweat pH and lactate, using three layers: (1) chitosan, (2) sodium carboxymethyl cellouse, and (3) indicator dye(methyl orange(MO)), bromocresol green (BCG). Reprinted with permission from [110]. Copyright 2018, Elsevier B.V. All rights reserved.

**Figure 12 sensors-22-00638-f012:**
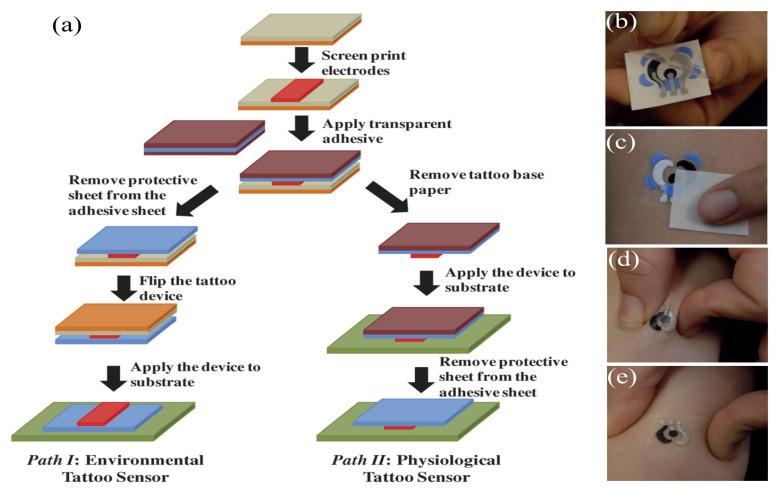
(**a**) The screen-printed electrochemically based tattoo, (Path I) for environment, (Path II) for analyte monitoring. Image (**b**) shows the clear protective sheet is removed, and in (**c**) the tattoo base paper is softly applied on human skin after dabbing with water for tattoo appliances. Image (**d**,**e**) illustrates the level of mechanical stress and strain that a tattoo is used to twist the underlying skin on a human subject. The figure concept adapted from [111,117].

**Figure 13 sensors-22-00638-f013:**
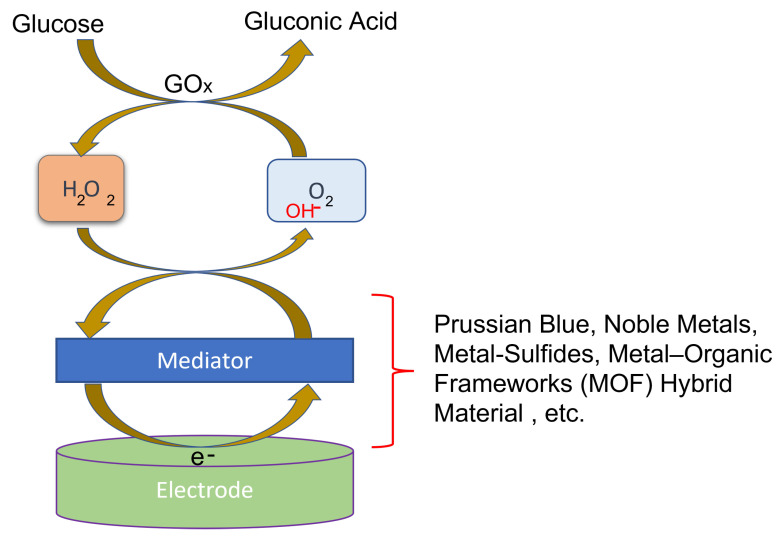
Schematic of a Enzymatic Chrono Amperometric biosensor with mediators, i.e., Prussian blue, nanozymes, etc., interacting with glucose oxidase to detect glucose in sweat. Adapted from [14].

**Figure 14 sensors-22-00638-f014:**
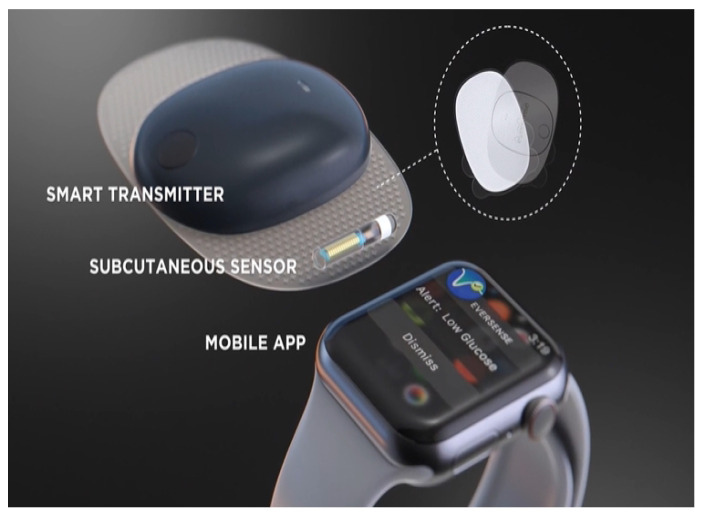
The “Eversense” CGM System from Ascensia Diabetes Care Group is a device that measures glucose levels utilizing sensor technology based on fluorescence: Image taken from https://www.ascensiadiabetes.com (accessed on 10 November 2021).

**Figure 15 sensors-22-00638-f015:**
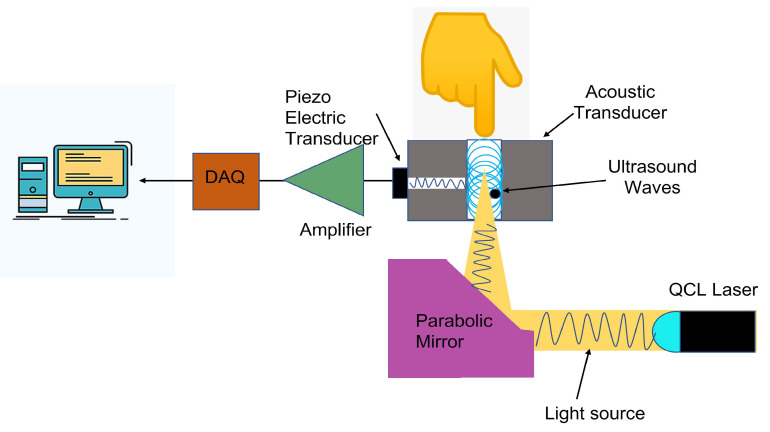
The basic configuration of photoacoustic spectroscopy (PAS) sensing is shown here. The laser beam from tunable and pulsed QCL laser is irradiated on skin, producing a thermal expansion in the skin to generate the ultrasonic wave. The thermal absorption and thus the ultrasound produced is dependent on ISF constituents such as glucose. The ultrasonic wave is amplified in the acoustic resonator, also known as the cell, and detected through a piezoelectric transducer such as a microphone. After that, the electrical signal corresponding to photoacoustic signal at the piezo sensor’s output is amplified, digitized, and forwarded to the computer for analysis. As the near- or mid-IR wavelengths are tuned one at a time, the amplitude of the corresponding photoacoustic signal indicates which wavelengths are absorbed more efficiently than others. Using ED-QCL, higher-order glucose wavelengths between 1 and 1.25 microns can be easily detected.

**Figure 16 sensors-22-00638-f016:**
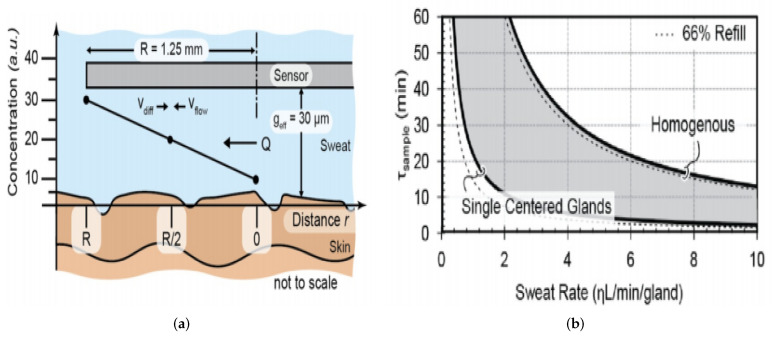
Effects of (**a**) the skin and sweat glands gap on (**b**) time necessary to fill the new sweat sample under a sensor. The figure concept adapted from [59].

**Figure 17 sensors-22-00638-f017:**
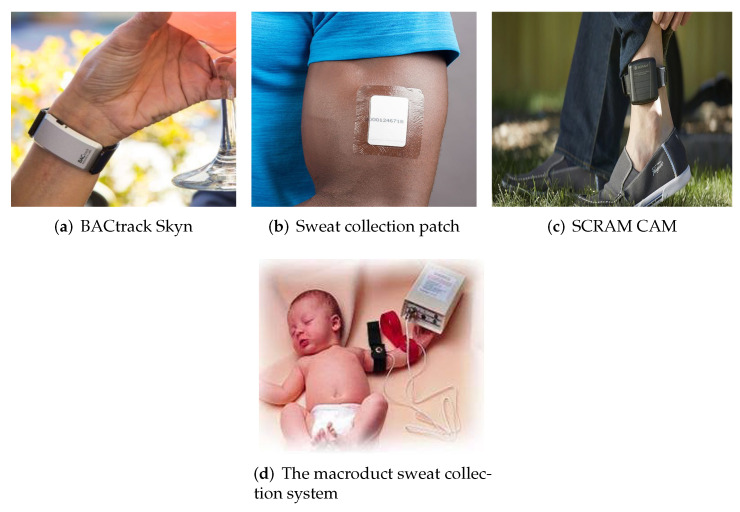
Wearable devices introduced by some companies that aspire to commercialize wearable sweat-sensing devices. (**a**) The BACtrack Skyn is a wearable alcohol monitor patch device for monitoring transdermal alcohol in real time. This product is for research only and is not yet commercially available; image taken from https://skyn.bactrack.com (accessed on 15 November 2021). (**b**) Sweat collection patch for drug testing developed and sold by Alcopro; image taken from https://www.alcopro.com (accessed on 15 November 2021). (**c**) The SCRAM CAM, an ankle bracelet for alcohol testing; image taken from https://www.scramsystems.com (accessed on 15 November 2021). (**d**) The ELITechGroup proposed a macroduct sweat collection system by ELITechGroup for diagnosis of cystic fibrosis; image taken from https://www.elitechgroup.com (accessed on 15 November 2021).

**Table 1 sensors-22-00638-t001:** Summarized studies on epidermal wearable glucose level monitoring systems.

Wearability	Biofluid Type	Sampling Method	Benefits	Next Steps	Reference
Eyeglasses sensor	Sweat	Exercise	Continuous monitoring of sweat glucose. Integration with wireless electronics.	Detail study with validation is required	[35]
Wearable patch with multimodal glucose sensor	Sweat	Exercise	Controlled sweat uptake. Improved accuracy of glucose sensing using multimodal sensing array and correction with sweat pH value.	Validation results required for continuous monitoring and replacement of commercial analyzer.	[36]
Graphene-based stretchable patch	Sweat	Exercise	Accurate monitoring by combination of pH, temperature, and humidity. Nanomaterials-based sensitive glucose sensor.	Increase sampling frequency and large-scale validation needed.	[37]
Wearable patch coupled with induced sweating	Sweat	Iontophoresis (stimulated)	Integration of iontophoresis sweat generation with glucose sensing.	Extension to on-body monitoring.	[38]
Multiplexed wearable, flexible array patch	Sweat	Exercise	Simultaneous multiplexed sweat sensing. Integration of customized wireless electronics.	Establish correlation to blood glucose, validation is also required.	[39]
Temporary tattoo	ISF	Reverse iontophoresis	Cost effective, easy to wear, and no skin irritation.	Single use, study stability, and reproducibility towards continuous use.	[33]
GlucoWatch	ISF	Reverse iontophoresis	FDA approved, provide continuous monitoring and electronics for measurement.	Minimize skin irritation, shorter warming up period, interference by sweat generation, and time lag compared to blood glucose.	[34]

**Table 2 sensors-22-00638-t002:** Summarized frequently used electrochemical methods for the detection of sweat in real time.

Methods	Overview	Advantages	Disadvantages	References
Potentiometry	The voltage from the sensor to the reference electrode indicates the ion concentration of the target.	Simple detection scheme and signal processing.	Since this system measures activity in contrast to concentration, selective membrane layers for certain ions must be developed.	[39,107,118,119]
Chronoam- perometry	The sensing electrode is subjected to a constant potential voltage, and the resulting current is proportional to the required analyte concentration due to the stimulation of redox processes.	Straight forward detection and post-processing. Redox Mediators can be employed for efficient electrons transfer and also to lower the necessary potential and thus power consumption (ferric materials are used as mediators such as ferrocene and Prussian blue).	In order for the enzymes to operate, a certain biochemical process must be initiated and stable under typical operating conditions of the biosensor. The design of the bio-sensor depends on both the knowledge of the target analyte and on the complexity of the analysis matrix.	[127]
Voltammetry	A voltage scan is performed between sensor and reference electrode, which extracts current properties for the concentration determination.	Having capability for analyzing two or more analytes in a single sample.	Low-detection level. The overlapping of redox potential, the presence of active intervening compounds and, the formation of intermetallic degradation-signal compounds contain significant problems.	[124,128]
Electrochemical Impedance Spectroscopy	The wearable EIS system is made up of a sensing analog front end that is combined with low-volume (1–5 L) ultra-sensitive flexible biosensors.	Provides more understanding of electrochemical system than any other electrochemical techniques. Covers a wide range of frequencies of the application samples.	Needs longer measuring and post-processing times to address greater uncertainties (8–10 h required).	[129]

**Table 3 sensors-22-00638-t003:** Summarized attributes for the potential market in different regions for wearable sweat glucose sensors according to the business reports from 2018–2027.

Attributes	Details
Base Year estimation	2018
Forecast period	2019–2027
Market Representation	Revenue in USD Million and CAGR from 2019 to 2027
Regional Scope	North America, Europe, Asia Pacific, Latin America, and MEA.
Country Scope	U.S., Canada, U.K., Germany, France, China, Japan, Brazil, Mexico
According to report [153]: sweat sensors are available by: Mechanism Sensing substance Material End User Geography.	Biochemical, Bioelectrical. Cortisol, Ethanol, etc. Soft Polymer, Plastic, others. Hospitals, Military, Sports. North America, Europe, Asia Pacific, Middle East, Africa, and Latin America.

## Data Availability

Not applicable.

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
