# Peer review of "Comprehensive Review on Wearable Sweat-Glucose Sensors for Continuous Glucose Monitoring"

_sensors, 2022, doi:10.3390/s22020638_

Round 1

Reviewer 1 Report

The paper by Zafar et al. deals with the Review on Wearable Sweat-Glucose Sensors.

-In my opinion, the paper suffers from clarity in the general presentation of the paper.

-The abstract seems not strictly connected with the  title and with the rest of the paper. 

However, this work is a good contribution to the field and could be published  after major revision as mentioned below:

  1. Abbreviation should be defined the first time used in the manuscript.
  2. Some figures including figure 5, 7 should be present in higher quallity.
  3. In figure 12, caption figure is not match with the figure. figure should show more ditails.
  4. In figure 9, there is no symetry proportion between A, and B,C,D, E.
  5. The title of section 2 is not related to its context! Basically, it seems there is no essential deffernt between section 2 and 3! they repeat each other. This review is based on wearable sensor but there is no a short meanwhile comprehensive definition of wearable (bio)sensors. It would be fine if a history of wearable biosensore added to section 2.
  6. In following of comment 5, section 2 is titled with the history of sweat glucose sensor, but figure 1 shows the fundamental principle of electrochemical biosensor. Authurs just implied that electrochemical sensors are the most commen techniques without explaining any spesification. It would be better to rewritten section 2 considering all mentioned points.

Author Response

Dear Reviewer,

Reviewer 2 Report

This manuscript systematically summarizes the sweat glucose monitoring, particularly the prospect of its commercialization. In addition, the challenges relating to sweat collection, sweat sample degradation, person-to-person sweat amount variation, various detection methods, and their glucose detection sensitivity, and also commercial viability are thoroughly covered. This review is systematic and comprehensive, especially the description of its prospect in commercialization. However, there are some problems needed to be emphasized before publication.

  1. The Introduction section is poorly written, and the motivation does not jump to the reader.

  1. The manuscript has an insufficient introduction to the glucose monitoring mechanism and should be described in more detail.

  1. The author should draw a mechanism picture to show the monitoring mechanism for glucose.

  1. The author should include a discussion on the stability of the reported Wearable Sweat-Glucose Sensors.

  1. There were some grammatical errors in the manuscript. And some words were misused.

  1. The reviewer suggests adding an extra section that discusses signal amplification of nanomaterials. Since some nanomaterials have enzyme-like activity and are usually called nanozymes, and they are used to interact with GOx for high-sensitive glucose sensing by signal amplification.

  1. Single-atom catalysts (SACs) with high peroxidase-like activity show massive potential in the wearable sensor field. It should be discussed in the perspective section. For example, some advanced Fe-N-C SACs can effectively electrochemical sensing H2O2 (Ding et al, 17 Small (2021): 2100664) and connect with GOx to detect glucose (Chen et al,Small 16.31 (2020): 2002343; Xiong et al, Science Bulletin 65.24 (2020): 2100-2106. )

Author Response

Dear Reviewer,

Round 2

Reviewer 1 Report

The revised version of the manuscript can now be accepted for publication.

Reviewer 2 Report

The revised version is good for me, I agree to accept it.